# The Role of Adiponectin and Leptin in Fibro-Calcific Aortic Valve Disease: A Systematic Review and Meta-Analysis

**DOI:** 10.3390/biomedicines12091977

**Published:** 2024-09-02

**Authors:** Veronika A. Myasoedova, Francesca Bertolini, Vincenza Valerio, Donato Moschetta, Ilaria Massaiu, Valentina Rusconi, Donato De Giorgi, Michele Ciccarelli, Valentina Parisi, Paolo Poggio

**Affiliations:** 1Centro Cardiologico Monzino IRCCS, 20138 Milan, Italy; francesca.bertolini@ccfm.it (F.B.); vincenza.valerio@ccfm.it (V.V.); donato.moschetta@ccfm.it (D.M.); ilaria.massaiu@ccfm.it (I.M.); valentina.rusconi@ccfm.it (V.R.); donato.degiorgi@ccfm.it (D.D.G.); 2Department of Medicine, Surgery and Dentistry, University of Salerno, 84084 Fisciano, Italy; mciccarelli@unisa.it; 3Department of Translational Medical Sciences, Federico II University, 80138 Naples, Italy; valentina.parisi@unina.it; 4Department of Biomedical, Surgical and Dental Sciences, University of Milan, 20122 Milan, Italy

**Keywords:** adipokines, adiponectin, leptin, fibro-calcific aortic valve disease, aortic stenosis

## Abstract

Background: Fibro-calcific aortic valve disease (FCAVD) is a progressive disorder characterized by the thickening and calcification of the aortic valve, eventually leading to aortic stenosis. Adiponectin and leptin, known for their anti-inflammatory and proinflammatory properties, respectively, have been implicated in cardiovascular diseases, but their associations with FCAVD are controversial. This meta-analysis aims to evaluate the relationships between adiponectin and leptin levels and FCAVD, particularly in patients with severe aortic stenosis (AS). Methods: A systematic search was conducted across the PubMed, Scopus, and Web of Science databases to identify studies on adiponectin and leptin levels in FCAVD. The methodological quality of each study was assessed using the Newcastle–Ottawa Scale. Standardized mean differences (SMDs) and 95% confidence intervals (CIs) were calculated, and publication bias was evaluated using Egger’s test and funnel plots. Results: Out of 191 articles identified, 10 studies involving 2360 patients (989 with FCAVD and 1371 controls) were included. The analysis suggested trends in the associations of lower adiponectin levels (SMD = −0.143, 95% CI: −0.344, 0.057, *p* = 0.161) and higher leptin levels (SMD = 0.175, 95% CI: −0.045, 0.395, *p* = 0.119) with FCAVD. The association remained a trend for low adiponectin but showed a significant correlation with high leptin in severe AS patients (SMD = 0.29, 95% CI: 0.036, 0.543, *p* = 0.025). Conclusion: This meta-analysis indicates a potential association between elevated leptin levels and severe aortic stenosis, while the relationship with adiponectin levels remains inconclusive. These findings highlight the need for further and dedicated research to clarify the roles of these adipokines in the pathogenesis of FCAVD and their potential roles as biomarkers for disease progression.

## 1. Introduction

Fibro-calcific aortic valve disease (FCAVD) is a progressive multifactorial disorder, initially characterized by the thickening and calcium accumulation of leaflets without hemodynamic changes, called aortic valve sclerosis, with the subsequent worsening of the disease with impaired hemodynamics, called aortic stenosis [1]. Previous studies have shown that FCAVD is associated with an increased risk of coronary artery disease, atherosclerotic disorders, heart failure, and cardiovascular and all-cause mortality [2]. Indeed, the pathophysiological processes involved in the development of FCAVD, such as inflammation, oxidative stress, endothelial dysfunction, lipid accumulation, and calcification, are similar to atherosclerosis [3]. 

In the quest to unravel the complexities of FCAVD, the role of adipokines emerges as a pivotal factor. Indeed, adiponectin and leptin, two hormones with opposing actions, have been implicated in the modulation of several cardiovascular pathologies. Adiponectin has been shown to have anti-inflammatory and insulin-sensitizing properties, as well as anti-atherogenic functions [4,5]. In coronary heart disease patients, adiponectin was positively associated with HDL cholesterol [6,7] and artery elasticity indices [7]. Accordingly, low adiponectin levels in non-diabetic and obese individuals were linked to intimal thickening, increased lipid-rich plaque, and elevated plasma lipoproteins, as well as insulin resistance [8]. In addition, in another study on CAD patients, the authors found an association between a decrease in adiponectin levels and an increment in inflammatory markers [9]. Meanwhile, in type 1 diabetic patients, lower plasma adiponectin levels were related to coronary artery calcification (CAC) [10]. Smoking, a risk factor for atherosclerosis, was associated with reduced adiponectin levels, potentially due to nicotine’s effect on adiponectin expression [11]. Furthermore, adiponectin was found to inhibit foam cell formation in macrophages by preventing lipid accumulation, highlighting its protective role against atherosclerosis [12]. In addition, an increase in valvular inflammation and faster aortic stenosis progression were found in patients with low adiponectin levels [13]. Indeed, low adiponectin levels were associated with AS in elderly patients [14].

In contrast, leptin has been shown to have proinflammatory properties, and increased circulating leptin levels were associated with cardiovascular disease and adverse outcomes [15,16]. In animal studies, leptin administration increased atherosclerosis and thrombosis, while leptin deficiency reduced atherogenesis [17,18]. Leptin was shown to upregulate certain proteins and signaling pathways in vascular endothelial cells, contributing to lesion development and smooth muscle cell proliferation [19,20]. Furthermore, aortic valve calcification was more pronounced under leptin treatment in an apolipoprotein e-deficient animal model [21] and highly expressed in human calcified aortic valve tissue [22]. A clinical study involving 174 men and 26 women with type 2 diabetes mellitus found a strong link between plasma leptin levels and coronary atherosclerosis [15]. However, the Multi-Ethnic Study of Atherosclerosis (MESA) found no correlation between leptin levels and cardiovascular events [23]. Nevertheless, a recently published study indicated that high leptin levels were associated with future AS surgery in patients with concomitant coronary artery disease [14].

The aim of our meta-analysis was to investigate the relationship between adiponectin and leptin levels and FCAVD, with a specific focus on patients with severe AS. Our objectives included examining the association between FCAVD and either a reduction in adiponectin or an increase in leptin levels across studies involving patients with various stages of FCAVD, ranging from aortic valve sclerosis to severe aortic valve stenosis. Additionally, we sought to determine if these two adipokines have a sex-specific effect on AS.

## 2. Materials and Methods 

### 2.1. Data Sources and Searches

A comprehensive protocol for the search strategy was prospectively developed, detailing the study’s objectives, selection criteria, outcomes, and statistical methods. This protocol was registered in the International Prospective Register of Systematic Reviews (PROSPERO: CRD42024562103) to ensure transparency and reproducibility.

To identify all relevant studies examining the association between adiponectin or leptin levels and fibrocalcific aortic valve disease (FCAVD), we conducted a systematic search in several electronic databases, such as PubMed, Scopus, and Web of Science. The search was performed in accordance with the Preferred Reporting Items for Systematic Reviews and Meta-Analyses (PRISMA) guidelines to ensure a thorough and unbiased selection of studies [24]. The search strategy included specific keywords and Boolean operators to capture a wide range of studies related to adiponectin, leptin, and FCAVD.

For the PubMed search, we used the following keywords: (adiponectin OR adipo OR adipoq OR adipokynes OR adipokine) AND ((aortic valve) OR (aortic valve disease) OR (calcific aortic valve) OR (aortic stenosis) OR (aortic sclerosis) OR (aortic valve sclerosis) OR (aortic valve stenosis) OR (CAVS) OR (CAVD) OR (AVSc)).

For the Scopus search, we used the following keywords: (adiponectin OR adipo OR adipoq OR adipokynes OR adipokine) AND ((aortic AND valve) OR (aortic AND valve AND disease) OR (calcific AND aortic AND valve) OR (aortic AND stenosis) OR (aortic AND sclerosis) OR (aortic AND valve AND sclerosis) OR (aortic AND valve AND stenosis) OR (CAVS) OR (CAVD) OR (AVSc)).

For the Web of Science search, we used the following keywords: (adiponectin OR adipo OR adipoq OR adipokynes OR adipokine) AND ((aortic valve) OR (aortic valve disease) OR (calcific aortic valve) OR (aortic stenosis) OR (aortic sclerosis) OR (aortic valve sclerosis) OR (aortic valve stenosis) OR (CAVS) OR (CAVD) OR (AVSc)).

The final search was completed in June 2024, ensuring that the most recent and relevant studies were included in the analysis.

### 2.2. Study Selection, Data Extraction, and Quality Assessment

In accordance with the established protocol, we included all studies that reported data on the association between adiponectin or leptin levels and FCAVD. We excluded case reports and reviews, as well as studies involving animal models. All the included studies were categorized based on the severity of aortic valve conditions: those involving subjects with aortic valve sclerosis and varying degrees of aortic valve stenosis (FCAVD) were analyzed separately from those focusing exclusively on severe aortic stenosis (AS).

For each included study, we extracted data on the key clinical and demographic characteristics of the patients with FCAVD and the control groups. The methodological quality of each study was assessed using the Newcastle–Ottawa Scale (NOS), which evaluates three main domains: selection, exposure, and outcome. The NOS scoring system ranges from 0 to 9, with higher scores indicating a better methodological quality (Appendix A).

Two independent operators (FB and VAM) conducted the data extraction and quality assessment. In cases of disagreement, a third operator (PP) was consulted to reach a consensus. Any discrepancies were resolved through discussion and mutual agreement.

### 2.3. Statistical Analysis and Risk of Bias Assessment

Statistical analyses were conducted using Comprehensive Meta-analysis Version 3.3.070 (Biostat, Englewood, NJ, USA, 2014). Differences in continuous variables were quantified as standardized mean differences (SMDs) with corresponding 95% confidence intervals (CIs). The overall effect size was evaluated using Z scores, with statistical significance set at *p* < 0.05.

To assess statistical heterogeneity among the included studies, we employed Cochran’s Q test and the I^2^ statistic. The I^2^ statistic quantifies the inconsistency across study results, indicating the proportion of total variation in study estimates attributable to heterogeneity rather than sampling error. Specifically, I^2^ values were interpreted as follows: 0% indicating no heterogeneity, 25% indicating low heterogeneity, 25–50% indicating moderate heterogeneity, and values exceeding 50% indicating high heterogeneity [25].

### 2.4. Publication Bias and Analytical Approach

Funnel plots of standard error by standard differences in means were drawn and visually inspected for asymmetry to detect potential publication bias. Additionally, Egger’s test was employed to quantitatively assess publication bias, with statistical significance set at *p* < 0.05. This objective evaluation complemented the subjective visual inspection [26]. In cases of significant publication bias, Duval and Tweedie’s trim-and-fill method was used to allow for the assessment of the adjusted effect size [27]. To account for variability among the included studies and ensure a robust analysis, we applied the random-effects model for all statistical analyses. This approach acknowledges and incorporates the heterogeneity across studies, providing more generalized and reliable results [28].

## 3. Results

### 3.1. Study Characteristics

The search strategy yielded 191 articles related to adipokines in FCAVD (Figure 1). After removing duplicates and screening titles and abstracts, twenty-eight articles were selected for full-text review. Upon further examination, 18 studies were excluded due to irrelevant content. Ultimately, ten studies [14,22,29,30,31,32,33,34,35,36] involving 2360 participants (989 FCAVD patients and 1371 controls) were included in the qualitative and quantitative analyses of adiponectin and leptin levels.

Data on adiponectin levels were reported in five studies [14,31,32,33,34] with a total of 1386 subjects (574 FCAVD patients and 812 controls), while eight studies [14,22,29,30,31,34,35,36] reported data on leptin levels in 2170 subjects (863 FCAVD patients and 1307 controls). In addition, four studies [14,31,33,34] showed adiponectin levels and five studies [14,22,29,31,35] leptin levels in patients with severe AS.

A total of 47% of the included patients were females, with a mean age of 66.5 years (range: 58–71 years; Appendix A). Overall, hypertension was presented in 73% of the included patients, diabetes in 31%, and coronary artery disease in 54%. The mean BMI was 27.5 kg/m^2^ (range: 24.9–31.4 kg/m^2^), the total cholesterol levels were 187 mg/dL (range: 151–242 mg/dL), the low-density lipoprotein cholesterol levels were 102 mg/dL (range: 79–117 mg/dL), the high-density lipoprotein cholesterol levels were 47 mg/dL (range: 43–51 mg/dL), and the triglyceride levels were 136 mg/dL (range: 122–150 mg/dL).

### 3.2. Adiponectin and Leptin Levels in FCAVD Patients

Differences in adiponectin and leptin levels in patients with FCAVD compared to controls are presented in Figure 2. The data suggest trends in the associations between lower adiponectin levels (SMD = −0.143, 95% CI: −0.344, 0.057, *p* = 0.161) and higher leptin levels (SMD = 0.175, 95% CI: −0.045, 0.395, *p* = 0.119) and FCAVD, although they were not statistically significant. 

The examination of the funnel plots of effect size vs. standard error, for the studies evaluating the difference in adiponectin and leptin levels between patients with FCAVD and controls, shows the absence of publication bias for adiponectin—confirmed by Egger’s test (*p* = 0.439)—and the presence of a publication bias (Egger’s *p* = 0.045; Appendix A). Thus, after adjusting for publication bias (Duval and Tweedie’s trim-and-fill analysis), these results were confirmed with an estimated point for the SMD of 0.38 (95% CI: 0.14, 0.62) for leptin levels.

### 3.3. Adiponectin and Leptin Levels in Severe as Patients

Analyzing the relationships between adiponectin levels in patients with severe AS only, we observed that lower adiponectin levels displayed a similar trend as with FCAVD (SMD = −0.256, 95% CI: −0.586, 0.074, *p* = 0.129; Figure 3A). However, the evaluation of leptin levels revealed that a higher leptin concentration was consistently linked with severe AS (SMD = 0.29, 95% CI: 0.036, 0.543, *p* = 0.025; Figure 3B). Of note is the fact that a high heterogeneity among the studies was observed for leptin in severe AS patient studies (I^2^: 71.2%, *p* = 0.008). Overall, the funnel plots of effect size vs. standard error for the studies evaluating the differences in adiponectin and leptin levels between patients with AS and controls show the absence of publication bias, confirmed by Egger’s tests for adiponectin and leptin (*p* = 0.978 and *p* = 0.157, respectively; Appendix A).

### 3.4. Sex-Specific Leptin Levels in as Patients

Remarkably, three studies [14,30,35] reported data on leptin levels in men and women separately (677 men and 594 women). A separate sensitivity analysis of these studies suggested that leptin levels were significantly lower in men compared to women (SMD = −3.438, 95% CI: −6.027, −0.849, *p* = 0.009), both in AS patients and in controls (Appendix A). However, high heterogeneity was also observed among the studies (I^2^: 99.4%, *p* < 0.001), with the absence of publication bias (*p* = 0.704; Appendix A).

## 4. Discussion

The results of our research indicate a modest association between fibro-calcific aortic valve disease and a decrease in adiponectin or an increase in leptin levels. However, considering only patients with severe aortic stenosis, a significant and pronounced correlation was noted with increased leptin levels. Notably, this relationship appears to be sex-specific, with variations in the leptin concentrations observed between men and women diagnosed with aortic stenosis.

Growing evidence from clinical studies and laboratory experiments highlights the numerous positive effects of adiponectin on cardiovascular disease [37]. Adiponectin plays a crucial role in cell–cell communication, being a mediator among adipose tissue, myocardial cells, and the vasculature [37]. Adiponectin is particularly considered a crucial factor in the connection between obesity and the onset of atherosclerosis [38]. In contrast to other adipokines, adiponectin has been described to have insulin-sensitizing, anti-inflammatory, and anti-thrombotic properties [39]. Indeed, a study by Mohty et al. [13] found that lower plasma adiponectin levels were linked to heightened valve inflammation and faster hemodynamic progression of aortic stenosis. This indicates that hypoadiponectinemia, often seen in obesity and metabolic syndrome, might contribute to the development of fibro-calcific aortic valve disease. In addition, adiponectin was shown to be inversely associated with aortic valve calcification (AVC) prevalence and extent [40]. In line with this hypothesis, it has been shown that lower adiponectin levels were associated with the earliest manifestation of aortic valve disease (i.e., aortic valve sclerosis) [32]. On the other hand, the results of another study showed that in AS with preserved ejection fraction, there were no changes in the profile of any adipokines compared with controls. In this study, adipokine levels were associated with coexisting atherosclerosis and not with typical cardiovascular risk factors or AS hemodynamic parameters [31]. Another recent study on this topic suggested that adiponectin was not associated with an increased risk of future aortic valve replacement (AVR) after 5 years of investigation before surgery; although, after stratification by age, higher levels of adiponectin were associated with a reduced risk of AVR in people aged ≥60 years at the time of surgery [14]. That being said, the results of our meta-analysis suggest that lower levels of adiponectin only have a tendency to correlate with fibro-calcific aortic valve disease or just severe aortic stenosis (i.e., the end-stage of aortic valve disease). 

Unlike adiponectin, leptin was shown to have the ability to stimulate platelet activation and smooth muscle cell proliferation as well as to induce inflammation, oxidative stress, and endothelial dysfunction [41,42]. High levels of leptin, or hyperleptinemia, may increase the risk of cardiovascular diseases [43,44] and contribute to the hardening of arteries due to its role in promoting the transformation of cells, which leads to vascular calcification [21]. Despite this, the Multi-Ethnic Study of Atherosclerosis (MESA) found that leptin levels were not linked to extra-coronary calcification [40]. Conversely, the results of another study showed that higher leptin levels were linked to the presence of fibro-calcific aortic valve disease, especially in patients under 65 years old and those with impaired renal function [36]. Our results are in line with the latter, since this meta-analysis revealed a consistent relationship between elevated leptin levels and severe aortic stenosis, even if, in the early stages, this association did not reach statistical significance. Of note, the direct association between leptin and severe AS could be explained through the effect that leptin has on isolated valve interstitial cells (VICs). Indeed, leptin could promote human VIC osteoblast differentiation in an Akt- and ERK-dependent manner [22].

Leptin represents a potential pharmacological target for cardiovascular disorders, including aortic valve disease. Recent research has highlighted that hyperleptinemia alone can induce leptin resistance [45]. Elevated leptin levels play a pivotal role in driving obesity and associated metabolic disturbances. Consequently, reducing circulating leptin levels emerges as an effective therapeutic strategy for managing obesity-related consequences [45]. The efficacy of partial leptin reduction, particularly in the context of hyperleptinemia, finds indirect support from various observations. Notably, the glucagon-like peptide-1 analog liraglutide significantly reduces cardiovascular events and mortality rates, which correlates with decreased leptin levels [46]. Additionally, inhibitors targeting the sodium–glucose-linked transporter-2 (SGLT-2) have emerged as potent cardiovascular drugs [47]. Indeed, the reduction in leptin levels due to SGLT-2 inhibition may contribute to their cardiovascular benefits [48], since hyperleptinemia leads to sodium retention and plasma volume expansion, triggering cardiac and renal inflammation as well as fibrosis. Furthermore, leptin-mediated neuro-hormonal activation appears to upregulate SGLT-2 expression in renal tubules [49]. Other potent cardiovascular interventions, such as antihypertensive therapy with perindopril [50], antidiabetic therapy with metformin, and lipid-lowering therapy with statins, have direct effects on adipocytes, particularly white adipose tissue, decreasing leptin expression [51,52]. Additionally, cannabinoid receptor 1 antagonists have been explored as an adjunctive strategy to reduce circulating leptin levels [53,54]. Zhao et al. [55] propose that directly targeting leptin neutralization, specifically in the context of hyperleptinemia, could lead to direct cardiovascular improvements. Indeed, leptin-neutralizing antibodies, which effectively lower circulating leptin levels, hold promise not only for weight loss but also for their anti-fibrotic and insulin-sensitizing effects [55]. However, further investigation is needed to understand the impact of these antibody-based approaches on the cardiovascular system.

Finally, leptin secretion exhibits sex-specific patterns [56] and, when elevated in conjunction with overweight and obesity, contributes to cardiovascular diseases in both sexes [57]. Our study’s findings are in line with these results, indicating that men have significantly lower leptin levels than women even in patients with severe AS. Typically, women produce three to four times more leptin than men, a difference attributed to women’s higher body fat percentage [58,59]. In the context of cardiovascular health, it has been established that leptin promotes hypertension through distinct mechanisms based on sex: it triggers sympathetic nervous system activation in men and stimulates the aldosterone–mineralocorticoid receptor pathway in women [60,61]. Considering the sex-specific characteristics of fibro-calcific aortic valve disease [62], and given that hypertension is a critical risk factor for AS development [63], the insights on this adipokine (i.e., leptin) could be instrumental in deepening our understanding of aortic valve pathophysiology. 

## 5. Limitations

This study has several limitations that should be considered when interpreting the results. First, the sample size is small, and the studies included are heterogeneous in terms of design and patient populations, which may affect the generalizability of the findings. Second, while the absence of publication bias was confirmed for adiponectin levels, a significant publication bias was detected for leptin levels, which could influence the reported associations. Although Duval and Tweedie’s trim-and-fill analysis adjusted for this bias, the potential impact on the study’s conclusions remains. Third, the cross-sectional nature of the included studies limits the ability to establish causality between adipokine levels and fibro-calcific aortic valve disease. Longitudinal studies are needed to better understand the temporal relationship and potential causal pathways. Lastly, the range of control groups in the included studies varied, and the control populations were not always well defined, which may introduce confounding factors and affect the validity of the comparisons made between patients with fibro-calcific aortic valve disease and controls.

## 6. Conclusions

Our meta-analysis showed that, while adiponectin and leptin are both implicated in cardiovascular pathologies, their roles in fibro-calcific aortic valve disease appear to be distinct. Our findings suggest that lower levels of adiponectin may be associated with the pathology, but this association is not as strong as the correlation observed with higher leptin levels, especially in patients with severe AS. Future studies should continue to explore (i) the sex-specific aspects of leptin’s effects to enhance our understanding and improve patient outcomes and (ii) the possibility of using leptin inhibitors or agents able to reduce its levels as a target for FCAVD.

## Figures and Tables

**Figure 1 biomedicines-12-01977-f001:**
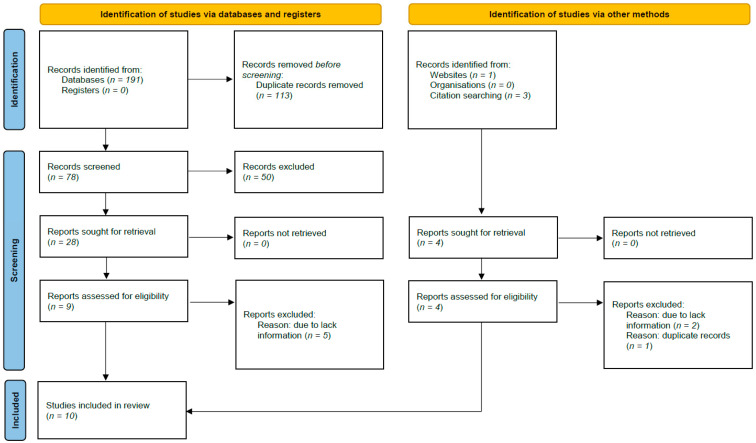
Prisma Flow Chart. The flow chart represents the number of studies evaluated according to PRISMA guidelines.

**Figure 2 biomedicines-12-01977-f002:**
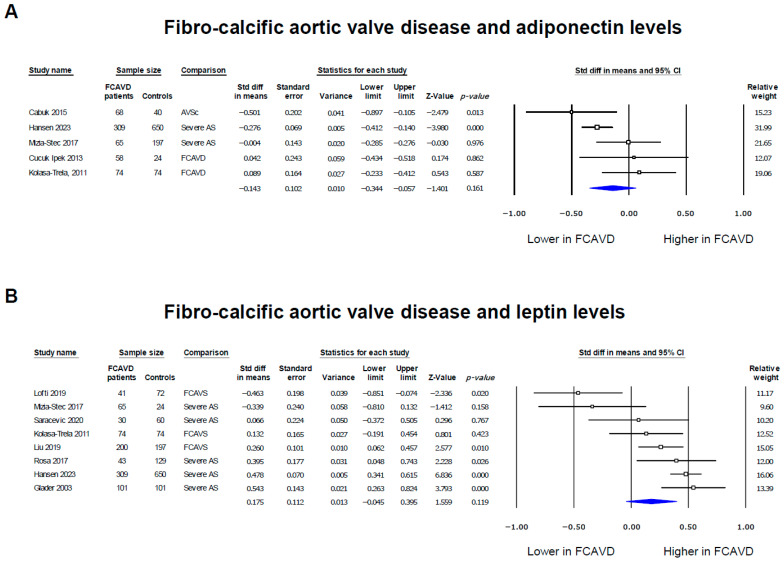
Forest plots of the adipokine levels’ associations with fibro-calcific aortic valve disease. Adiponectin [14,31,32,33,34] (**A**) and leptin [14,22,29,30,31,34,35,36] (**B**) levels in patients with fibro-calcific aortic valve disease were presented as standardized differences in means (SMDs) compared to controls. The blue diamonds represent the estimated overall effect, while the squares represent each study with 95% CIs.

**Figure 3 biomedicines-12-01977-f003:**
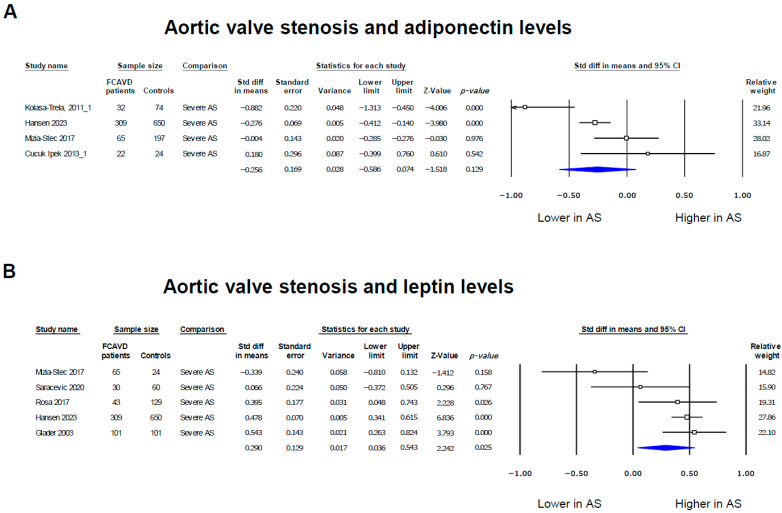
Forest plots of the adipokine levels’ associations with severe aortic valve stenosis. Adiponectin [14,31,33,34] (**A**) and leptin [14,22,29,31,35] (**B**) levels in patients with aortic stenosis were presented as standardized differences in means (SMDs) compared to controls. The blue diamonds represent the estimated overall effect, while the squares represent each study with 95% CIs.

## Data Availability

The data supporting this article will be available upon reasonable request to the corresponding author.

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
