# Peer review of "The Role of Adiponectin and Leptin in Fibro-Calcific Aortic Valve Disease: A Systematic Review and Meta-Analysis"

_biomedicines, 2024, doi:10.3390/biomedicines12091977_

Round 1

Reviewer 1 Report

Comments and Suggestions for Authors

The review by author emphasizes the Contrasting Roles of Adipokines in Fibro-Calcific Aortic Valve Disease. The manuscript could be improved by addressing the following comments:

·       Authors are suggested to improve the abstract section aligning the overall theme of the review.

·       Add more content related to the role of adipokines and rationale of the review in the introduction section.

·       Authors are suggested to improve and provide more recent content related to the present work. There is very limited content in the introduction section.

·       Result section must be divided into subsections either on the basis of various roles of adipokines or by disease condition wise.

·       Authors are suggested to include good schematic diagrams depicting mechanism of action of adipokines.

·       Add table to represent the various studies of adipokines and their mechanism of action.

·       Improve the conclusion section by incorporating the futuristic approach related to the therapeutic approach of the adipokines in  Fibro-Calcific Aortic Valve Disease.

·       Authors are suggested to format the MS as per the journal’s guidelines.

·       Check the whole manuscript for possible typos and grammatical errors.

Comments on the Quality of English Language

Minor corrections are required

Reviewer 2 Report

Comments and Suggestions for Authors

I read with interest the paper by Veronika A. Myasoedova et al.

The authors attempted to shed some more light in the implication of adipokines in the processes of fibrocalcific aortic valve disease in humans.

I have a few comments and remarks.

This meta-analysis is well-conducted and the manuscript is well-written  with appropriate utilization of robust analytical and statistical tools. 

There are only a few grammatical errors, eg, page 2, first line: 'disfunction'.

The authors acknowledge the limitations of the study which are clearly noted in the corresponding section, primarily the heterogeneity of the different cohorts studied which may have an impact on the generalizability of the observations. My main concern is that the authors combined studies that in their totality included patients all across the spectrum of fibrocalcific disease and different degrees of hemodynamic impairment, from aortic valve sclerosis to severe aortic valve stenosis. Nonetheless, common pathophysiologic pathways may dominate the disease across the spectrum. Interestingly though, this was not the case, but rather, only high leptin levels were found to correlate statistically with only severe aortic stenosis.

In any case, I believe that there is additional value in the content of this study and hence, overall, the content will be of relative interest to the readers and I would support the publication of the article.

Comments on the Quality of English Language

Only minor grammatical errors identified. Overall, well-written manuscript.

Round 2

Reviewer 1 Report

Comments and Suggestions for Authors

Authors have incorporated all the suggested changes hence the manuscript can be accepted in its current form.

Author Response

Comment 1: Authors have incorporated all the suggested changes hence the manuscript can be accepted in its current form.

Response 1: We thank the Reviewer for the positive feedback. We are pleased to hear that the manuscript has met expectations.